# Variables and Mechanisms Affecting Response to Language Treatment in Multilingual People with Aphasia

**DOI:** 10.3390/bs10090144

**Published:** 2020-09-22

**Authors:** Mira Goral, Aviva Lerman

**Affiliations:** 1Speech-Language-Hearing Sciences, Graduate Center & Lehman College, City University of New York, New York, NY 10016, USA; 2MultiLing Center for Multilingualism in Society across the Lifespan, University of Oslo, 0313 Oslo, Norway; 3Program of Communication Disorders, Hadassah Academic College, Jerusalem 9101001, Israel; alerman@gradcenter.cuny.edu

**Keywords:** language activation, language inhibition, language treatment, cross-language generalization, stroke aphasia

## Abstract

Background: Despite substantial literature exploring language treatment effects in multilingual people with aphasia (PWA), inconsistent results reported across studies make it difficult to draw firm conclusions. Methods: We highlight and illustrate variables that have been implicated in affecting cross-language treatment effects in multilingual PWA. Main contribution: We argue that opposing effects of activation and inhibition across languages, influenced by pertinent variables, such as age of language acquisition, patterns of language use, and treatment-related factors, contribute to the complex picture that has emerged from current studies of treatment in multilingual PWA. We propose a new integrated model—Treatment Effects in Aphasia in Multilingual people (the TEAM model)—to capture this complexity.

## 1. Introduction

This paper addresses the variables and mechanisms that affect efficacy of language treatment provided for multilingual people who acquire language impairments (aphasia). Our aim is to shed light on a complex phenomenon and to reflect on the mixed patterns of findings regarding cross-language response to intervention reported to date. We begin with a brief overview of aphasia generally and aphasia in multilingual people more specifically. We then review and discuss the variables that have been implicated in affecting the patterns of improvement among multilingual individuals and that contribute to levels of language activation and inhibition that have been proposed to account for the patterns observed. We do not endeavor to present an exhaustive review of the literature on intervention in multilingual people with aphasia. Instead, we set out to highlight the variables that have been discussed in this literature as factors that affect cross-language treatment generalization. Furthermore, we put forward a model that incorporates the factors that need to be considered in understanding intervention outcomes in multilingual individuals with aphasia. This model highlights the mechanisms that have been previously proposed to explain cross-language generalization and suggests how pertinent variables may affect these mechanisms.

Aphasia is an acquired language disorder resulting from injury to the brain, most typically due to a stroke. People with aphasia experience difficulty with language production and comprehension. These difficulties range from mild to severe, and may affect all language modalities (e.g., speaking and reading), and one or more linguistic ability (morphology and semantics). The language impairment experienced by people with aphasia (PWA) affects their ability to communicate and thus their engagement in family, social, daily-living, and work environments. PWA typically improve their abilities over time, especially with intervention directed toward restoring the impaired abilities and also toward compensating for the deficits. However, complete recovery from aphasia is rare, e.g., [1].

Multilingual people, that is, people who use more than one language regularly [2], who acquire aphasia typically exhibit impairment in all their languages, to similar or varying degrees. Research reports on multilingual PWA have documented that a small majority of individuals experience comparable difficulties in all their languages (parallel impairment), provided they had pre-stroke high proficiency in these languages, or experience comparable impairments relative to the degree of language proficiency and use they had prior to the aphasia onset. In some cases, however, differential degrees of impairment of one or more languages are noted (non-parallel impairment) [3,4,5]. For example, in highly proficient bilingual people, one language can be impaired, while their other language remains relatively intact post-stroke [6]. Several variables have been implicated in determining patterns of impairment and recovery in multilingual PWA. These include variables related to the type of multilingualism, such as age of language acquisition, relative pre-stroke proficiencies of each language, and frequency of language use and exposure, as well as variables related to the stroke, such as site and size of the brain lesion and the time elapsed since the aphasia onset [7,8,9].

Intervention in aphasia has been shown to improve language comprehension and production and to facilitate recovery, e.g., [10]. PWA of varying severity and profiles exhibit common characteristics (e.g., word finding difficulties, termed *anomia*), but also a range of heterogeneous impairments and heterogeneous response to intervention. Researchers and clinicians have examined factors that influence the degree to which PWA respond to intervention and have been working toward identifying variables that can predict best response to therapy [10,11,12]. With multilingual PWA, in addition to the greater heterogeneity among them than among monolingual PWA (including varying degree of language abilities prior to the aphasia onset), decisions have to be made regarding which language(s) to select for the intervention. The literature on treatment effects in multilingual PWA is particularly mixed in terms of approaches used and findings reported, e.g., [13,14,15].

The mechanisms underlying treatment-related improvement in aphasia have been studied in recent years, but much is still not well understood [16,17]. The evidence suggests that neuroplasticity allows for reorganization of portions of the neuronal networks, and that regions adjacent to or homologous to the affected regions take on language-related processing [18,19]. Similar processes can be assumed to account for changes observed in multilingual PWA, but an additional consideration is required. Namely, expectations regarding changes associated with treatment would depend on hypotheses regarding the neuronal networks that are associated with the processing of multiple languages in the brain.

Much has been debated about the underlying networks subserving multiple languages. For example, brain stimulation studies [20] and some fMRI studies [21] have demonstrated differential patterns of activation for the different languages of a multilingual person during language processing. In contrast, other neuroimaging investigations have demonstrated complete overlap in the patterns of activation during fMRI studies of multiple languages [22,23]. Moreover, Abutalebi and Green [24] argue that even when there are differences in the patterns of activation during language processing observed with neuroimaging, these can be attributed to mechanisms of activation and inhibition of the relevant languages rather than to the processing of the languages per se. Researchers who put forward mechanisms to account for the complex picture of improvement in multilingual PWA draw on what has been proposed in the study of multilingual language processing. Psycholinguistic investigations of language performance in multilingual individuals suggest that, in the presence of multiple languages, mechanisms of activation as well as mechanisms of inhibition regulate language use, depending on the communication situation [25].

During language production, words and structures from all languages of multilingual speakers are presumed to be activated. When the conversation context is multilingual, they may mix words and structures from different languages within the conversation (termed code switching) and also within a conversation turn (termed code mixing) [26,27]. If the conversation context requires the use of only one language, mechanisms of language inhibition allow the speakers to avoid language mixing. Inhibition can take place at different stages of language production—while formulating the message or selecting the sentence structure and lexical items, during the retrieval of the words, and during the selection of the word form to be produced, e.g., [28,29]. Multilingual PWA may have specific deficits in their languages, as well as impairments of the activation and inhibition mechanisms, affecting access to their languages [30]. These linguistic and mechanistic deficits can interact with a number of variables to determine the response of multilingual PWA to intervention.

In Section 2, Section 3 and Section 4 below, we highlight examples from the literature on treatment in multilingual PWA, illustrating variables that may account for patterns of cross-language response to language intervention. To this end, we discuss nine key variables that have dominated the literature on cross-language treatment in multilingual PWA: three multilingualism-related variables (age of language acquisition, language use and exposure, language proficiency pre stroke); two stroke-related variables (brain lesion site, time post-onset); language abilities post-stroke, a variable that combines multilingualism-related and stroke-related effects; two treatment-related variables (focus of treatment, language of treatment); and linguistic distance, a multilingualism-related variable that interacts with treatment focus. The aim of this paper is not to provide an exhaustive review of the literature on cross-language generalization in aphasia; rather, we set out to highlight the variables and mechanisms that have been discussed in previous publications but may have not been considered in concert. To illustrate each variable, we selected one example from the published literature that demonstrates the role of the given variable in the outcomes observed. In Section 5, we discuss two key mechanisms that have been proposed in the literature to account for the interplay among the languages of multilingual PWA in response to treatment. We then present an integrated model of Treatment Effects in Aphasia in Multilingual people (the TEAM model) of the variables reviewed in Section 2, Section 3 and Section 4 that we argue affect these mechanisms and, in turn, response to treatment. A brief conclusion is offered in Section 6 and future directions are suggested in Section 7.

## 2. Multilingualism-Related Variables Affecting Response to Treatment in Multilingual PWA

In this section, we highlight four key variables that are unique to multilingual people and have been discussed in the literature on cross-language generalization (three are purely multilingualism-related and one is related to multilingualism as well as to stroke, as discussed below). The first—age of language acquisition—is inherent to each multilingual person and is unchangeable. The following two—language use and exposure, and language proficiency pre-stroke—are dynamic, inter-related, and typically change across the multilingual lifespan. The fourth—post-stroke language abilities—encompasses the joint effects of language proficiency pre-stroke and the effect of the acquired deficits post-stroke. One other multilingualism-related variable is the linguistic distance (similarities and differences) among the languages acquired and learned by multilingual speakers. However, we address this variable in the section on treatment below (see Section 4.1), in keeping with the focus of the literature on linguistic distance and treatment effects.

### 2.1. Age of Language Acquisition

Multilingual people may be exposed to two or more languages from birth, develop high proficiency in those languages, and use all languages extensively. Examples of such simultaneous multilinguals include people who are born in multilingual societies (e.g., Spanish–English communities in the US; Catalan–Spanish in Catalonia, Spain; English–Afrikaans–Xhosa in Cape Town, South Africa; French–English in Montréal, Canada). Whereas many childhood multilingual people maintain balanced, high proficiency in both or all their languages, in some cases, especially when the immediate language environment (e.g., in the immediate family) is different from the larger sociolinguistic context (e.g., the community, the country), the language used by the majority group in the larger social context becomes the more dominant language of the individual (e.g., English for Spanish–English bilingual people in the U.S.).

Other multilingual individuals may acquire one language from birth—their L1—and a second or additional languages in early childhood (early sequential bilingualism) or in late childhood or adulthood (late sequential bilingualism). Numerous variations of such circumstances have been recorded: people who are born into families that use a language different from the majority language in the environment; people who are born in one language environment and move to another in childhood or adulthood; people who live with speakers of other languages; and so forth. In these scenarios, there may be a difference in the representation and processing of the first-acquired language, the L1, versus any later learned languages, e.g., [31]. For example, research has suggested that memory systems associated with a later learned language may be explicit as compared to the implicit learning associated with a first-acquired language [5,31]. As well, learning a language by immersion characterizes the acquisition of a first language and early learned languages, as compared to greater reliance on metalinguistic skills that are typically involved during learning a second language later in life [5].

Studies that examine the special status of an L1 in aphasia have suggested that the first-acquired language of multilingual PWA is likely to be less impaired as compared to later learned languages [8], but there are also examples when this is not the case, e.g., [3,32]. In the literature on language intervention with multilingual PWA, several studies provide evidence that supports differential processing of an L1 and non-L1s, including differential response to treatment in the L1 and in a non-L1 [14,33,34] as evident in the following example.

Miertsch et al. [34] examined treatment effects in a German–English–French trilingual participant with aphasia. The participant, who sustained a stroke at age 48, reported high proficiency and frequent use of his three languages. In the study, the treatment was administered in the participant’s L3, which he learned in late childhood. The treatment targeted language production and comprehension, focusing on word finding of verbs, nouns, and prepositions in isolation and in discourse context, and on semantic-conceptual relationships between words. The assessment comprised the Bilingual Aphasia Test (BAT) [35], an aphasia battery that assesses linguistic abilities (phonology, morphology, syntax, lexicon, and semantics) across all linguistic modalities (listening, speaking, reading, and writing). The authors found that following treatment, the participant improved his test scores in the treated language, French, as well as in his English, which was not treated. There was no significant improvement in the participant’s performance in his L1, German. The lack of cross-language treatment benefit to the first-acquired language can also be explained, however, by the higher language abilities the participant demonstrated in his L1 post-stroke compared to his other two languages.

### 2.2. Language Use and Exposure

Frequency of language use typically correlates positively with language proficiency. However, the frequency and habits of language use can have an independent effect on the degree of language activation and thus of its accessibility. At one extreme, a language that is used most post-aphasia onset will likely improve more with treatment. On the other end of the continuum, people who do not use a language at all in the months and years prior to or following aphasia onset may experience language attrition that compounds the effect of the aphasia and renders the activation of that language more demanding.

Language attrition has been most studied in the context of immigration in which individuals move to a language environment that limits their exposure to and use of one of their languages. L1 attrition—the reduced accessibly of an L1 following immersion in a later learned non-L1 and the disuse of L1—has been studied extensively, e.g., [36,37,38]. The literature on L1 attrition reveals that people who stop using their L1, typically due to immersion in their L2, experience gradual difficulties in accessing the language. The complete or partial reduction in use of and exposure to the language typically affects production language abilities, such as word retrieval and idiom use, before affecting comprehensions abilities [37,38,39]. L1 attrition rarely results in a complete loss of the language, especially if the acquisition had been complete before the change in the language environment. Attrition of other languages has also been studied, albeit to a lesser degree [40,41]. Decreased lexical retrieval abilities is a hallmark of the reduced accessibility associated with language attrition, in both L1 and L2. In multilingual PWA, reduced use of a language, prior to as well as following the aphasia onset, can contribute to reduced responsiveness to language treatment [32,42]. Findings from these studies demonstrate that despite language attrition, treatment can be effective, although cross-language generalization from treatment in a more-used language to a language that has undergone attrition has not been found.

Several treatment studies converge to suggest that even in the absence of attrition, treatment in multilingual PWA may be most efficacious in the language most used in the environment [42,43,44]. For example, Goral and colleagues [43] examined treatment effects in a multilingual PWA who was highly proficient in his two early acquired languages (Spanish and Catalan), and less proficient in three additional languages (English, French, and German). The participant, who sustained a stroke at age 44, lived in Spain where he was mostly exposed to Spanish and Catalan. The study documents the treatment administered to the participant while he visited New York City for several months. Following treatment in Spanish, the participant made little improvement; following treatment in English—the language of the environment at the time of the study—the participant demonstrated substantial gains in his language production (as measured by picture naming, verbal fluency, elicited sentence production, and semi-spontaneous speech) in English, as well as in his other languages.

Language use can be affected by the language of the immediate environment (e.g., family) or extended environment (e.g., the community and the country). It can also be influenced by affective variables such as attitude toward the language and toward its speakers, e.g., [3], the status of the language in the society, e.g., [7], and other psychosocial factors, such as incentives to use a language of vocation.

### 2.3. Language Proficiency

Whereas any language difficulties exhibited by monolingual speakers who acquire aphasia can be attributed to their aphasia (providing childhood language development deficits and other neurological or cognitive disorders are ruled out), this is not necessarily the case for multilingual PWA. Multilingual individuals vary in the degree of their language proficiency. Some people have high, native-like proficiency in multiple languages, e.g., [45], but more commonly, multilingual people report having higher proficiency in one language over the others. In many instances, the language of higher proficiency is also the first acquired, L1. However, it is not uncommon that a later learned language becomes the more proficient or more dominant language, especially for people who move from their L1 language environment to an environment where their later learned language is the dominant language.

Measuring language proficiency in multilingual speakers is not a trivial matter. There are formal foreign language proficiency tests, such as The Test of English as a Foreign Language and The European Consortium for the Certificate of Attainment in Modern Languages, which can be helpful in determining proficiency levels, but this kind of information is virtually never available for PWA about their abilities prior to their aphasia onset. Instead, the assessment of pre-stroke language proficiency of multilingual PWA has to rely on subjective self-ratings and on family members’ report. This subjective information is problematic for two reasons. One, it has been demonstrated that self-reported proficiency and objective language testing correlate positively but weakly [46,47]. Two, language proficiency is a dynamic construct that may change for individuals throughout their lifetime and with changes in their living environments and sociolinguistic contexts. Asking people to rate their abilities will thus be a very time-consuming process or a crude approximation. Researchers studying multilingual speakers with aphasia continue to improve the field’s understanding of the reliability of self-reported proficiency, the wording that best elicits the target information, and the inter-relation between language proficiency and language use [9].

A language that achieved only low or moderate proficiency before the stroke may appear inaccessible to a person with aphasia. In several studies in the literature with PWA who knew multiple language before the stroke, assessment and treatment were administered only in the more proficient languages of the participants. Additional languages, which the participants reported knowing but with a low degree of proficiency and use, are not typically assessed and are rarely treated, e.g., [33,48]. An example of such a choice is reported in Goral, Levy, and Kastl [33]. The participant enrolled reported high pre-stroke proficiency in Hebrew, English, and French, and these three languages were assessed prior to and following treatment, which was provided in English, the participant’s second language. The participant also reported working knowledge in three additional languages: Spanish, German, and Italian, but those languages were not formally assessed in the study as the participant reported minimal access to those languages following his stroke.

### 2.4. Post-Stroke Language Abilities

Determining language abilities in PWA after the aphasia onset is challenging as well, because observed performance on language tests inherently reflects the combined effects of language proficiency (pre-stroke) and aphasia severity, which are difficult to dissociate. Multilingual PWA who report comparable as well as differential proficiency levels prior to the aphasia onset have been included in treatment studies. In many of those cases, the same relative abilities were evident prior to the stroke (as estimated) and following the stroke (as measured), but some changes in language dominance post-stroke have also been reported. Mixed results found to date make it challenging to ascertain differential effects of relative pre-stroke proficiency and post-stroke abilities on cross-language treatment generalization [43,49,50,51].

The effects of post-stroke abilities can be appreciated in the following example. Conner and colleagues [49] enrolled in a treatment study a multilingual PWA who had high proficiency pre-stroke and good abilities post-stroke in several languages, and lower abilities post-stroke in several others. The authors were able to examine the role of post-stroke levels of language abilities in the effects observed following treatment administered in the participant’s L1. The participant was a 64-years-old man who acquired Dutch (Flemish) as his L1, and was exposed to Dutch, French, and German—all spoken in his environment—from early childhood. Throughout his adulthood he learned English, Danish, Swedish, Norwegian, Italian, Spanish, and Portuguese, and has used them with varying degree for his professional and social interactions. His self-rated proficiency in these languages indicated high post-stroke abilities in five languages (Dutch, English, French, German, and Italian) and somewhat lower post-stroke abilities in other languages. Following intervention in his L1, which targeted efficiency of language production, the authors observed gains in the treated language, as measured by the proportions of words that contributed to a meaningful production of the total words produced, in a variety of elicited production tasks (e.g., picture-based description and answering wh-questions). Gains were also evident in each of his four languages with high abilities comparable to the L1, albeit to varying degrees, but were absent from the two weaker languages examined (Spanish and Norwegian).

We note again that language abilities post-stroke is a multilingualism-related variable that represents the joint effects of the proficiency levels attained in a language prior to the stroke as well as the effect of the acquired impairments due to the stroke. The next section focuses on stroke-related variables.

## 3. Stroke-Related Variables Affecting Response to Treatment in Multilingual PWA

### 3.1. Lesion Site

The severity of the aphasia and the characteristics of the impairments observed in people who acquire aphasia following a stroke have been associated with the site and size of the brain lesion, although there are no linear or one-to-one associations that would allow accurate predictions of the aphasia based on the location of the lesion, e.g., [52,53,54]. In most instances, selective damage to the left hemisphere results in aphasia and this has been found for monolingual as well as for multilingual individuals. Within frameworks that postulate dissociations in the representation of different languages in the multilingual brain, a differential response to treatment may be expected. Moreover, treatment in one language may not necessarily affect the untreated language(s). However, within neurolinguistic theories that postulate overlapping language representation in the brain and separate networks associated with controlling the activation of each language, treatment may be expected to positively affect all languages regardless of the language of treatment, unless the control mechanisms are affected by the brain lesion. In this latter case, one language may be less accessible and thus less responsive to treatment than the others.

Cross-language treatment effects have been reported for a substantial number of multilingual PWA who vary in their lesion size and site, which makes drawing clear conclusions about the role of the lesion site in determining response to treatment challenging. Nevertheless, several treatment studies have proposed evidence for impaired control mechanisms and consequent effects on response to treatment [55,56,57].

For example, Abutalebi et al. [55] enrolled a 56-years-old Spanish–Italian bilingual speaker in a treatment study. His stroke affected the lenticular nucleus of his left basal ganglia, which has been hypothesized to be critical to language control mechanisms [24]. Treatment, focusing on word retrieval, was administered in the participant’s L2, Italian, for six weeks and assessment was conducted in both languages, using the BAT [35] and a picture naming test. The authors reported improvement in the treated language and increased connectivity between the activation in the treated language and the language control networks; there was no change in the language that was not treated. Abutalebi and his colleagues [55] implicated the site of lesion, which included portions of the basal ganglia, in the failure to activate the participant’s L1, which resulted in the absence of cross-language treatment generalization.

### 3.2. Time Post-Onset

Positive response to treatment in aphasia has been demonstrated in various patients and to varying degrees. One variable that has been examined is the time elapsed since the aphasia onset. Greater neuroplasticity has been documented in the first weeks and months following the stroke (the acute and sub-acute phases) as compared to a longer period post-onset (the chronic phase) [58]. Nevertheless, treatment-related change has also been observed in chronic patients. It is possible that treatment-related changes observed in the acute phase are indicative of reorganization in the neuronal networks adjacent to the lesion site and these are likely to affect all languages, at least within a framework that postulates overlapping representation of multiple languages. Treatment-related changes in the language abilities of PWA in the chronic phase may be associated with reorganization of the linguistic system and with the development of compensation strategies, which may be directly related to the treatment provided. Here therefore one might expect restricted generalization from the language of treatment to the other language(s), perhaps depending on the target of the treatment (see Section 4.1 below). Most studies of aphasia treatment in multilingual individuals with aphasia have been conducted with people in the chronic phase. A few, however, report on treatment with PWA in the first weeks following the stroke, e.g., [44,55,59]. The following example illustrates the potential role of time post-onset in the observed treatment effects in a multilingual PWA.

Gil and Goral [59] discuss the challenge of determining the degree to which improved performance following treatment in the acute/subacute phase can be attributed to the treatment administered or to processes of spontaneous recovery that have been documented to take place in the first weeks following a stroke. The participant in their study, who was a 57-years-old Russian–Hebrew speaking man, was assessed two weeks after his stroke and then started to receive treatment. The first treatment was administered in Hebrew, the participant’s later learned second language. Assessment after a month of treatment revealed improved abilities in both the treated L2 and the untreated L1. Assessment after an additional two months of treatment in L2 revealed continued improvement in both languages but greater effects were observed in the untreated L1. Assessment after six weeks of treatment in L1 that followed revealed gains in both languages. At the time of the stroke the participant was exposed to and used both languages, living in Israel in a bilingual community. The greater response of his L1 to the treatment administered in L2 may reflect the interaction of spontaneous recovery and treatment-related recovery.

## 4. Treatment

Aphasia treatment includes a variety of approaches to linguistic and communication rehabilitation which vary in their focus and dose. Treatment can be geared toward restituting the impaired linguistic ability (e.g., word retrieval difficulty and reduced syntactic structures) as well as toward compensating for the impairment (e.g., using circumlocutions and using supported conversation) [60]. Not only the target of the intervention, but also its intensity and schedule may vary, factors that have been shown to potentially affect the efficacy of the treatment [61,62,63]. The efficacy of aphasia treatment has been measured in intervention studies with PWA by examining direct treatment effects, that is, improvement of the linguistic skills or linguistic stimuli that were targeted and practiced during the treatment sessions. Efficacy is also measured by examining the generalization of treatment effects beyond the specific tasks and stimuli practiced during the sessions, which is the ultimate goal of any intervention with PWA. Generalization can be measured by tests and tasks that examine language and communication behaviors not directly practiced during the sessions, and this includes both in the treated language (within-language generalization) and in the untreated language(s) (cross-language generalization). The findings concerning cross-language generalization in multilingual PWA suggest that two additional variables play a role. One is the focus of the treatment: the linguistic or communication aspects targeted during the treatment can determine whether cross-language generalization following treatment is observed, according to the linguistic distance between the two languages in question. That is, the degree to which the aspects that are targeted in the treatment are similar or different across the treated language and the other languages is likely to affect the degree to which cross-language generalization will occur. Two, the language selected for treatment can affect the degree to which cross-language generalization is observed. We turn to these two variables next.

### 4.1. Treatment Focus

By and large, treatment studies targeting the retrieval of specific lexical items have found that multilingual PWA improved their lexical retrieval of the trained items as well as of the translation equivalents of those items in the untreated language [50,64], although some individuals do not show this cross-language benefit [51]. Models of multilingual lexical representation, such as the Revised Hierarchical Model [65] and the Bilingual Interactive Activation Model [66,67] predict the spreading of activation along the semantic network of multiple languages (as will be discussed further in Section 5 below), which would account for the generalization of treatment effects from trained items in one language to their equivalents across languages. Language-specific variables, such as word frequency and word imageability that may affect processing at the word level have not been explicitly addressed in the literature on cross-language generalization. Multilingual-specific factors, especially cognate status (i.e., the degree of overlap in form and meaning between two translation equivalents), have been the focus on several investigations, yielding mixed results [68,69,70].

Studies that have employed treatment that targeted syntactic aspects of language production or processing in multilingual PWA are sparse; findings suggest that aspects that are shared among languages are more likely to exhibit cross-language benefits than aspects that differ across the relevant languages. For example, Goral and colleagues [33] administered intervention in their participant’s L2 English, and examined cross-language generalization to the participant’s L3 French and to his L1 Hebrew. The participant had acquired Hebrew from birth, English in early childhood, and learned French in adulthood, and reported high proficiency in all three languages prior to a stroke he sustained at age 42. Following the stroke, he experienced greater difficulties in English and French as compared to Hebrew. When treated in English, for his morphological and syntactic impairments, the participant showed improvements in the treated language, as measured by a picture-based connected speech elicitation task. Improvements included higher rates of morphosyntactic accuracy (e.g., noun–verb agreement) and increased speech rate. Improvement was also noted on similar measures in French, which was not treated. These effects were particularly notable for morphosyntactic components that are shared between English and French (e.g., pronoun–gender agreement) but not for those that diverge in the two languages (e.g., determiner–noun agreement).

### 4.2. Language of Treatment

For multilingual people who acquire aphasia, treatment choices mentioned above include an additional one, namely, the selection of the language of treatment. Virtually all studies of treatment with multilingual PWA examined the effect of treatment in one language per treatment block. Much attention has been paid to the relation between the relative abilities in the treated language compared to the other language(s) (see Section 2 above) and the likelihood of cross-language generalization to the language(s) that were not treated [42,49,64]. Studies have found cross-language positive effects between two languages of comparable abilities, e.g., [50,71,72]. When the language of treatment exhibits markedly stronger or markedly weaker abilities than the untreated language(s), mixed results have been reported, e.g., [14,49]. The leading theory that has been introduced in the literature for cross-language effects between languages of unequal abilities, namely, language control, is addressed in Section 5 below.

One study that included a bilingual treatment block in addition to single language blocks found that the bilingual treatment was not particularly efficacious [70]. In the study, the authors enrolled a Spanish–English bilingual participant with severe aphasia. The BAT [35] was used to assess her skills in both languages prior to and following treatment. The treatment was designed to target word production and to promote spreading activation across the semantic network, and was administered in three consecutive blocks: Spanish only, English only, mixed Spanish and English. The results demonstrated that the mixed treatment block was the least successful of the three. Additional preliminary results corroborate the finding that encouraging participants to mix their language within a treatment session may not be beneficial, showing minimal gains in either language [71].

## 5. Mechanisms Accounting for Observed Response to Treatment in Multilingual PWA

As illustrated in Section 2, Section 3 and Section 4 above, researchers of treatment in multilingual PWA have reported a mixed pattern of results and have identified critical variables that may influence the results observed. The effects of such variables can be viewed in the context of the mechanisms that have been proposed to underline how the languages of a multilingual speaker facilitate activation of one another as well as to interfere with and inhibit one another.

### 5.1. Activation

One account of the language difficulties experienced by PWA is that the levels of language activation are depressed and therefore greater activation of a linguistic component (e.g., a word) is needed for it to reach its threshold for selection, e.g., [73,74]. When multiple languages are involved, a given language may have to reach a threshold level of activation to be accessible for comprehension and production [5], although it has also been hypothesized that in a neurologically intact brain, all languages are always active [28,29]. The brain lesion resulting in language impairments in multilingual PWA can lead to depressed activation levels in all languages, or, it is possible that only one of the languages reaches the activation threshold at a given time, yielding the patterns of selective impairments that have been reported in the literature [5]. When treatment is administered to multilingual PWA to facilitate increased language activation, the activation can be specific to the treated aspects of the treated language (e.g., practiced lexical items), or it can spread to other aspects of that language (within-language generalization to unpracticed lexical items and to unpracticed tasks). If the spreading activation reaches the language that was not treated, cross-language generalization is observed [64]. This notion of spreading activation has been discussed in the context of semantic memory in early studies e.g., [75,76], and more recently also in the context of aphasia rehabilitation in multilingual people [64]. Whereas the focus of the current literature on spreading activation in multilingual individuals with aphasia has been at the word level, there is the potential to expand these theories to other aspects of language, similar to the development of bilingual language representation models that also branch out from the single-word level [77]. It may be expected that activation can spread across languages in any shared components, such as morphology and syntax, as indicated in Section 4.1 above.

As mentioned above, several studies with multilingual PWA have demonstrated that the benefits observed in the treated language generalized to the other languages, which were not treated [50,51,56,78]. For example, Lerman and colleagues [56] explained the improvements they observed in the treated and untreated languages of their participants by assuming a spreading activation within the semantic network following treatment that targeted strengthening connections within the network. Their participants were multilingual speakers who were born in an English-speaking environment (L1), acquired Hebrew from elementary school-age, and lived in a Hebrew-speaking environment at the time of their stroke. Both participants received treatment that targeted their underlying semantic network (Verb Network Strengthening Treatment: [79]). Following treatment in the participants’ L1 English, the authors observed improvement in that language, as measured by a variety of language production tasks of words, sentences, and connected language production. Improvements were also documented on similar measures in Hebrew, the participants’ later learned language, after treatment in English. The authors interpreted their results as evidence that due to the strengthening of connections within the semantic network following the VNeST, word retrieval improved, and, due to both the shared semantic network across languages and the presumed inter-connectivity of lexical items in the two languages, this improvement was observed in both languages.

The increased activation of the treated language can also result in increased interference from that language when attempts are made to access the untreated language(s). Such interference can prevent cross-language treatment generalization and could lead to decreased performance in the untreated languages at post-treatment assessment. For the spreading activation to successfully decrease the threshold for all languages and lead to cross-language benefits, the language control mechanism needs to be functioning properly, that is, applying the appropriate levels of inhibition depending on the communication situation [25]. Furthermore, processes of spreading activation need to be sufficiently strong to overcome processes of interference and inhibition [56,64]. Stronger spreading activation may be expected for languages of comparable levels of proficiency, for those languages that continue to be used by the multilingual speakers and in their environment, as well as across structures or features that are shared across the languages.

### 5.2. Inhibition

In addition to mechanisms of activation responsible for the ability to communicate in one or more languages, mechanisms of inhibition have been theorized to allow seamless language selection in neurologically healthy multilingual individuals [80]. In healthy multilingual brains, these language control mechanisms regulate levels of activation and inhibition of each language; in multilingual PWA, the brain lesion may impair these mechanisms, potentially preventing sufficient activation or resulting in excessive inhibition of one of the person’s languages. Treatment that is provided in one language of a multilingual PWA may increase the activation levels of that language, but might also increase the inhibition levels of the untreated, non-target language. Increased inhibition may also occur when interference from the untreated language needs to be controlled [80,81]. This may be particularly expected when the treated language is a weaker language, for which to achieve sufficient activation, increased inhibition of the stronger, more dominant, potentially interfering language needs to be applied. In this case, following treatment, lingering suppression of the untreated language could result in apparent negative cross-language effects [32,56].

For example, in the Lerman et al. study described in Section 5.1, cross-language benefits were observed following treatment from the participants’ L1 English to their later acquired Hebrew, but following treatment in the participants’ post-stroke weaker Hebrew, improvement was observed only in the treated language. Whereas one participant did not show change in his L1 English following treatment in Hebrew, the other participant exhibited a negative effect of treatment in Hebrew on his English during the testing immediately following the treatment. These effects can be accounted for by processes of inhibition of the stronger language (English) that linger following the treatment in the weaker language (Hebrew). There is some evidence that these effects are transient and that when activation in the stronger language resumes after treatment in the weaker language ends, activation of the stronger language returns to former levels [56].

### 5.3. Interactions of Activation and Inhibition and the Variables that Affect Them

Levels of activation and inhibition in the languages of multilingual PWA are influenced by the variables outlined in Section 2, Section 3 and Section 4 above. For example, recent use of a language and exposure to it in the sociolinguistic context would increase the activation of that language, whereas language attrition due to lack of exposure and use would decrease the activation levels of that language. Similarly, treatment that targets aspects that are shared among languages may increase the activation levels of all the languages as compared to treatment that targets language-specific characteristics which may facilitate those language aspects only in the treated language. This complex interaction of a number of such variables could be the reason for the plethora of mixed results reported in the literature on language treatment in multilingual PWA. We propose the TEAM model that considers critical aspects relevant to response to treatment in multilingual PWA (see Figure 1).

As can be seen in the model, the observed cross-language treatment effects (lower third of the model) are dependent on the interaction between activation mechanisms and inhibition mechanisms (middle third of the model), as discussed in Section 5. The relative impact of these activation and inhibition mechanisms is illustrated by the thickness of the arrows leading to each potential cross-language treatment effect: positive, null, or negative. The variables affecting each of the mechanisms (upper third of the model) can be split into three main groups.

First are the multilingualism-related variables, illustrated in the model by ellipses. One is fixed from the onset of bilingualism: age of acquisition (see Section 2.1); two are dynamic across the lifespan: language use and exposure (see Section 2.2) and pre-stroke language abilities (see Section 2.3). These multilingualism-related variables will directly affect activation and inhibition mechanisms (the dot-dash arrows in the model). For example, activation levels are expected to be higher in the language of the environment than another language [43], and lower in an attrited language [82]. Inhibition of an attrited language is also expected to be stronger than a language which is used frequently [82].

Second are the stroke-related variables, which include two variables: the brain lesion site (see Section 3.1) and time post-onset (see Section 3.2), illustrated in the model by the shape of an explosion. The lesion can affect the language network (i.e., the brain regions, tracts, and subcortical matter most associated with language see [53,83]) or the language control network [24], or both. As the arrows in the model indicate, the language network is expected to be most responsible for appropriate functioning of activation mechanisms and the language control network is expected to be most responsible for appropriate functioning of inhibition mechanisms. However, these two networks are not separate; not only do they partially overlap anatomically [24], they also work in conjunction with each other when they are unimpaired by a lesion. Time post-onset will affect the lesion in terms of neuroplasticity, either due to spontaneous recovery or due to the effects of treatment over time [16,17,59].

An additional variable is post-stroke language abilities (see Section 2.4), which are influenced by multilingualism-related variables (see Section 2) as well as by the stroke (see Section 3). This post-stroke language abilities variable is illustrated in the model as a star (combining the ellipse and explosion shapes). The other multilingualism-related variables have been suggested to affect post-stroke language abilities of multilingual people with aphasia, either directly or indirectly, as indicated by the solid line arrows in the model. Indeed, post-stroke language abilities are not necessarily fixed, but can be dynamic relative to time post-stroke and to language use and exposure after the stroke. Post-stroke language abilities affect activation and inhibition mechanisms, such that a less impaired language will have higher activation levels than a more impaired language [56,64]. Furthermore, in order to successfully process information in a more impaired language, the less impaired language will need to be strongly inhibited at that time [32,56]. Thus, post-stroke language abilities strongly influence activation and inhibition mechanisms and resulting treatment effects. Other bidirectional interactions between multilingualism-related variables and stroke-related variables also exist. For example, the more time post-onset that passes with no or minimal use of one language (due to differential language impairment, change in language of environment, etc.) the more that language is expected to undergo attrition.

A third group of variables is the treatment-related variables, illustrated in the model by squares. These variables can be manipulated, as opposed to multilingualism-related and stroke-related variables, which cannot usually be manipulated. Effective treatment would potentially increase activation levels of all languages (treated and untreated) while limiting any increase of interference (the dot-dash arrows in the model), allowing for activation mechanisms to be stronger than inhibition mechanisms [64,84]. In order to boost the efficacy of treatment, the language of treatment must be carefully selected, since it will influence the activation and inhibition mechanisms. Furthermore, the focus of the treatment (e.g., using cognates or non-cognates) will affect both activation levels and potential interference that must be controlled (see Section 5). Relatedly, linguistic distance is a multilingualism-related variable but one that has been discussed in the treatment literature most directly as a variable that interacts with the focus of treatment (see Section 4.1); it is therefore depicted in the model in a shape combining an ellipse and a square.

The specific constellation of these nine variables is unique to each multilingual individual with aphasia, resulting in the individual’s response to aphasia treatment.

## 6. Conclusions

In this paper we highlighted and illustrated variables that have been found to affect treatment outcomes in multilingual people with aphasia. We argue that these variables work in tandem to yield the complex set of results that has been reported in the literature. Findings from published studies suggest that age of acquisition, relative levels of language proficiency and of language use and exposure, and processes of language attrition contribute to cross-language treatment efficacy. In addition, characteristics of the brain lesion causing the aphasia, as well as the focus and language of the treatment administered, have been associated with the outcome documented for multilingual PWA. Furthermore, these variables modulate processes of language activation and inhibition that have been postulated to account for the single or mixed language selection that multilingual people exercise. These very processes are regulated via mechanisms of language control, which may be impaired in aphasia and may influence response to treatment in one language or more.

## 7. Limitations and Future Directions

Although we did not execute an exhaustive or systematic review of the literature, we focused on mechanisms and variables that have emerged in recent years in the discussion of cross-language generalization of aphasia treatment. The examples of studies we provided were chosen to illustrate the potential effects of each variable discussed.

In building on the mechanisms that have been put forward in the literature [56,64], we presented the TEAM model that attempts to capture the combined effects of key multilingualism-related, stroke-related, and treatment-related variables on those mechanisms. The model can serve as a road map for researchers and clinicians as they consider treatment and cross-language effects in multilingual people with aphasia. Although a direct testing of the model may prove difficult given the heterogeneity inherent to this population, the model represents a first step toward a better understanding of the diversity of cross-language treatment effects in multilingual people with aphasia. Future directions include a comprehensive review of the existing body of literature on cross-language generalization to examine the relative weight of these variables. In addition, computational models that attempt to estimate the effects of individual variables on the outcomes observed, such as the one developed by Kiran, Peñaloza, and colleagues, represent a productive way to improve clinical practice in this population [12].

## Figures and Tables

**Figure 1 behavsci-10-00144-f001:**
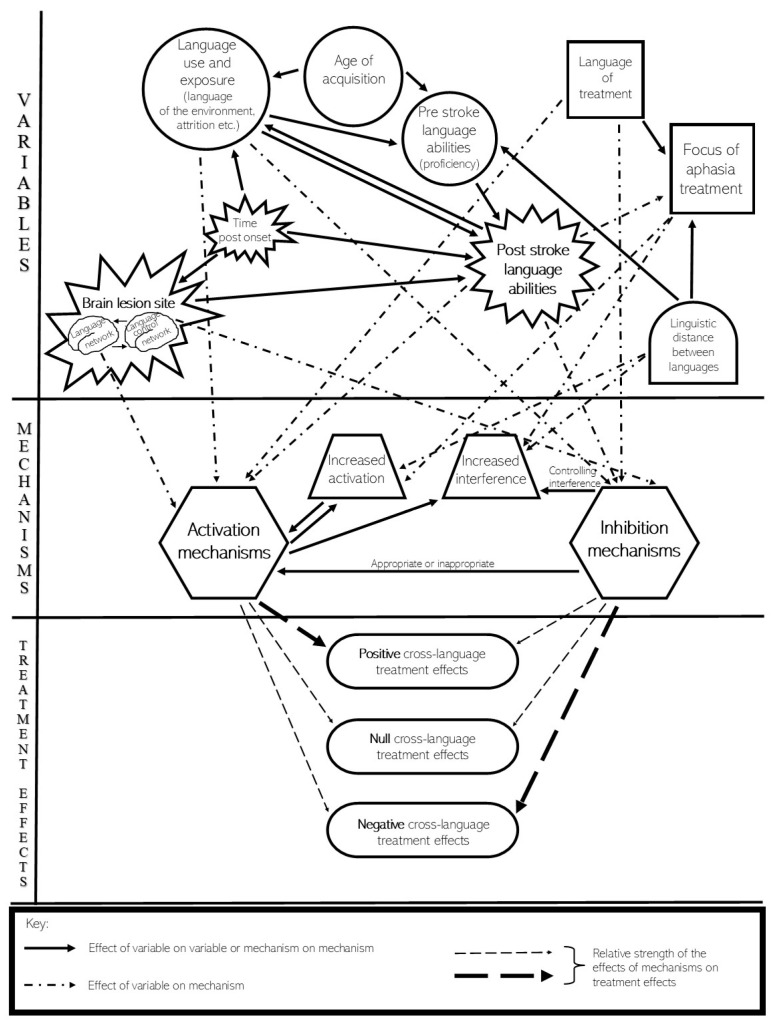
An integrated model of Treatment Effects in Aphasia in Multilingual people (TEAM) depicting the interaction of variables and mechanisms affecting cross-language effects following treatment in multilingual people with aphasia.

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
