# Peer review of "Variables and Mechanisms Affecting Response to Language Treatment in Multilingual People with Aphasia"

_behavsci, 2020, doi:10.3390/bs10090144_

Round 1

Reviewer 1 Report

The authors propose an integrated model (TEAM) of treatment effects in multilingual aphasia, distinguishing between multilingualism and aphasia variables, and arguing that the mechanisms of activation and inhibition within and across languages are key to understanding inconsistency among research findings. The multilingualism variables reviewed in the literature are: age of language acquisition, relative pre-stroke proficiencies of each language, and frequency of language use; the aphasia variables are: lesion site and size and the time elapsed since aphasia onset. The purpose of the TEAM model is stated as: "to aid further study of impairment and rehabilitation in multilingual people with aphasia".                                                       

The model is both ambitious and a welcome addition to the field. My remarks, some non-conventional for a review, are an attempt to make an already good paper better. I focus on two issues: spreading activation and the manner in which the model depicts the variables and mechanisms.

Activation is presented as one of two mechanisms which mediates the variables and the treatment effects. Spreading activation is introduced as a notion to account for generalization of treatment effects in multilingual aphasia research. This is a promising approach, and I believe new for the field of multilingual aphasia. The authors cite Kroll's Revised Hierarchical Model and Dijkstra's BIA+ model as their theoretical and empirical bases. Parenthetically, it may be useful for the readership of BehSci to include a reference to some of the classic papers in spreading activitation (e.g. Loftus 1973; Collins & Loftus 1975).

The claim is that treatment effects generalize "from trained/practiced to untrained/unpracticed items and tasks within the same language as well as across languages. It would be useful to specify in greater detail how exactly activation operates, i.e. how does activation spread? from concepts/words to larger units of language?  Beyond lexical activation, can we talk about activation of structures, of a language in general? It would be worth speculating on spreading activation beyond semantic networks, e.g. can we talk about activation of pragmatics, e.g. conversational routines, politeness strategies? How does spreading activation work in in the non-treated language? Does it differ from within language spreading activation?

In this context, I include an additional comment:

lines 360-363

"For the spreading activation to successfully decrease the threshold for all languages and lead to cross-language benefits, the language control mechanism needs to be functioning properly what constitutes proper functioning?. Furthermore, processes of spreading activation to overcome processes of interference and inhibition [49, 53]. suggest some ways of keeping activation strong

Review of the structure and claims of TEAM

Given the large number of connecting arrows and their differentiation in terms of direction, type of effect (within variables/mechanisms and between variables and mechanisms), and strength of effects, I scrutinized the model carefully in order to understand its claims. I did this both from top to bottom and from left to right.

Variables. Inspecting the model from top to bottom, it is divided into three horizontal sections. The top level contains the multilingual and aphasia variables, the middle section the activation and inhibition mechanisms, and the bottom layer the treatment effects. At the top, seven multilingualism variables are represented by ellipses: 'language of the environment,' 'language use and exposure,' and 'age of acquisition' run along the top, 'attrition' and 'pre stroke language abilities' below these, and 'time post stroke' immediately below attrition. All these variables are contained in ellipses of the same size. A seventh variable, 'post stroke language abilities' is contained in a larger ellipsis. Four of these variables are more closely related to multilingualism than the other three, which are stroke-related. The authors might consider distinguishing these two clusters in the text as well as in the diagram.

Brain lesion. To the left of the above-mentioned variables (ellipses), a jagged edged ovoid depicts the Brain Lesion itself. It contains two interacting brain hemispheres, one for the language network and one for the language control network. Three arrows come out of this representation, two to the Activation and Inhibition mechanisms in the middle layer of the model, and one to the variable labeled 'post stroke language abilities.' The text should specify the functions of the language and language control network.

Aphasia variables. To the far right of the top level of the model are the Aphasia variables, represented by rectangles of various sizes. At the center is 'Aphasia Treatment' in the largest rectangle, surrounded by 'Language of Treatment,' 'Dose,' and 'Focus of Treatment,' all with arrows entering the Aphasia Treatment variable. One additional variable is depicted in a rectangle with an arrow entering the 'Focus of Treatment' variable immediately above.

The model is far more detailed than the prose section of the literature review in the paper, which contains only three multilingualism variables and two aphasia variables. This difference should be noted.

Mechanisms. The middle and lower levels of the TEAM model are both symmetrical and thus easier to read and interpret, despite the large number of arrows (11) entering from the variable level above. Since the activation and inhibition mechanisms are the central explanatory constructs in the model and since one of the stated purposes of the model is to organize the large number of studies in a single framework, I would devote additional prose to referencing those papers which have documented evidence for activation and inhibition (perhaps in footnotes below the model).

Treatment effects. The bottom section of the model makes a tri-partite division of Treatment Effects into positive, null and negative effects with cross-language and within-language effects contained in each module. Six arrows come from the mechanisms above, one to each of the three possible treatment effects. The two strongest effects are depicted by bolded black lines, one from Activation to the positive treatment effects and one from Inhibition to the negative treatment effects. Here I make two suggestions: First, I would separate graphically within-language from cross-language effects, even at the expense of adding more complexity to the model. Second, it would be beneficial to further explicate the nature of the Activation and Inhibition mechanisms that lead to those effects, differentiating those which are within language and cross language.

Finally, I offer comments on a number of local issues:

lines 69-73

"Namely, expectations regarding changes associated with treatment would depend on hypotheses regarding the neuronal networks that are associated with the representation and processing of multiple languages in the brain."I would opt for focus on selection and control rather than on representation to sharpen this (see Green&Abutalevi 2008)

lines 81-86

"Researchers who put forwardàpropose mechanisms to account for the complex picture of improvement in multilingual PWA draw on what has been proposed in the study of multilingual language processing. Psycholinguistic investigations [add reference] of language performance in multilingual individuals suggest that, in the presence of multiple languages, mechanisms of activation as well as mechanisms of inhibition regulate language use, depending on the communication situation.

line 102

put forwardàproposed

line 182

that contributed to a meaningful production out of the total words produced, in a à

that contributed to meaningful production of the total number of words produced in a

line 184

languages with comparably high abilities to the L1,à

languages with high abilities comparable to those in L1,

line 190

On the one extreme à At one extreme

line 192

prior to or following the aphasia onset à delete 'the'

lines 322-328

"As illustrated in sections 2 and 3 above, researchers of treatment in multilingual PWA have reported a mixed pattern of results and have identified critical variables that may influence the results observed. The effects of such variables can be explained by postulating the underlying mechanisms that allow for the languages of a multilingual speaker to facilitate activation of one another as well as to interfere with and inhibit one another. I'm not sure a model which lacks the kind of detail that for example Green's language control model has is sufficient to 'explain' the effects of the variables proposed in the TEAM model. Green 2018 argues that "Varying language activation levels is an insufficient mechanism to explain the variety of language use [in CS]". I recommend a softer term.  

lines 333-338

"When multiple languages are involved, a given language has to reach a threshold level of activation to be accessible for comprehension and production [5]. The brain lesion resulting in language impairments in multilingual PWA can lead to depressed activation levels in all languages, or, it is possible that only one of the languages reaches the activation threshold at a given time, yielding the patterns of selective impairments that have been reported in the literature [5]."

Reconsider the notion of threshold level of activation; perhaps add a caveat.

lines 397-399

This complex interaction of a number of such variables can explain the plethora of mixed results reported in the literature on language treatment in multilingual PWA.

Consider changing 'explain' to 'organize,' 'clarify' or something more modest; see comment above.

Author Response

We thank the reviewers for their helpful comments. We revised the manuscript according to the reviewers’ suggestions. We simplified our model and added an explanation of the model’s structure. We wish to clarify that we did not aim to conduct a systematic review of the literature, and we now state so in the paper. Our aim was to highlight the variables that have been addressed in the literature on cross-language generalization in aphasia in multilingual people and to propose a model that can serve as a roadmap for researchers so key variables and mechanisms can be considered in tandem. In the revised manuscript, we made sure that the variables discussed in the text align with those included in the model and we added a detailed explanation of the model. We kept all new text in “track changes” as suggested.

Point-by-point response:

The authors propose an integrated model (TEAM) of treatment effects in multilingual aphasia, distinguishing between multilingualism and aphasia variables, and arguing that the mechanisms of activation and inhibition within and across languages are key to understanding inconsistency among research findings. The multilingualism variables reviewed in the literature are: age of language acquisition, relative pre-stroke proficiencies of each language, and frequency of language use; the aphasia variables are: lesion site and size and the time elapsed since aphasia onset. The purpose of the TEAM model is stated as: "to aid further study of impairment and rehabilitation in multilingual people with aphasia".                                                       

The model is both ambitious and a welcome addition to the field. My remarks, some non-conventional for a review, are an attempt to make an already good paper better. I focus on two issues: spreading activation and the manner in which the model depicts the variables and mechanisms.

Activation is presented as one of two mechanisms which mediates the variables and the treatment effects. Spreading activation is introduced as a notion to account for generalization of treatment effects in multilingual aphasia research. This is a promising approach, and I believe new for the field of multilingual aphasia. The authors cite Kroll's Revised Hierarchical Model and Dijkstra's BIA+ model as their theoretical and empirical bases. Parenthetically, it may be useful for the readership of BehSci to include a reference to some of the classic papers in spreading activitation (e.g. Loftus 1973; Collins & Loftus 1975).

Reply: We add reference to some of the classic papers on spreading activation as suggested (p. 9)

The claim is that treatment effects generalize "from trained/practiced to untrained/unpracticed items and tasks within the same language as well as across languages. It would be useful to specify in greater detail how exactly activation operates, i.e. how does activation spread? from concepts/words to larger units of language?  Beyond lexical activation, can we talk about activation of structures, of a language in general? It would be worth speculating on spreading activation beyond semantic networks, e.g. can we talk about activation of pragmatics, e.g. conversational routines, politeness strategies? How does spreading activation work in in the non-treated language? Does it differ from within language spreading activation?

Reply: we add a few sentences speculating about the scope of the spreading activation as suggested by the reviewer (p. 9)

In this context, I include an additional comment:

lines 360-363

"For the spreading activation to successfully decrease the threshold for all languages and lead to cross-language benefits, the language control mechanism needs to be functioning properly what constitutes proper functioning?. Furthermore, processes of spreading activation to overcome processes of interference and inhibition [49, 53]. suggest some ways of keeping activation strong

Reply: We added a sentence to clarify what may constitute proper functioning and one speculating about predicted stronger spreading activation (pp. 9-10).

Review of the structure and claims of TEAM

Given the large number of connecting arrows and their differentiation in terms of direction, type of effect (within variables/mechanisms and between variables and mechanisms), and strength of effects, I scrutinized the model carefully in order to understand its claims. I did this both from top to bottom and from left to right.

Variables. Inspecting the model from top to bottom, it is divided into three horizontal sections. The top level contains the multilingual and aphasia variables, the middle section the activation and inhibition mechanisms, and the bottom layer the treatment effects. At the top, seven multilingualism variables are represented by ellipses: 'language of the environment,' 'language use and exposure,' and 'age of acquisition' run along the top, 'attrition' and 'pre stroke language abilities' below these, and 'time post stroke' immediately below attrition. All these variables are contained in ellipses of the same size. A seventh variable, 'post stroke language abilities' is contained in a larger ellipsis. Four of these variables are more closely related to multilingualism than the other three, which are stroke-related. The authors might consider distinguishing these two clusters in the text as well as in the diagram.

Reply: We revised the model and distinguished the clusters of variables by shape; we now provide a detailed explanation of the model on pp. 12-14.

Brain lesion. To the left of the above-mentioned variables (ellipses), a jagged edged ovoid depicts the Brain Lesion itself. It contains two interacting brain hemispheres, one for the language network and one for the language control network. Three arrows come out of this representation, two to the Activation and Inhibition mechanisms in the middle layer of the model, and one to the variable labeled 'post stroke language abilities.' The text should specify the functions of the language and language control network.

Aphasia variables. To the far right of the top level of the model are the Aphasia variables, represented by rectangles of various sizes. At the center is 'Aphasia Treatment' in the largest rectangle, surrounded by 'Language of Treatment,' 'Dose,' and 'Focus of Treatment,' all with arrows entering the Aphasia Treatment variable. One additional variable is depicted in a rectangle with an arrow entering the 'Focus of Treatment' variable immediately above.

The model is far more detailed than the prose section of the literature review in the paper, which contains only three multilingualism variables and two aphasia variables. This difference should be noted.

Reply: We thank the reviewer for this suggestion and added details in the text accompanying the model. We revised the way we divide the variables throughout the paper and made sure that the model includes the variables addressed in the sections of the paper.

Mechanisms. The middle and lower levels of the TEAM model are both symmetrical and thus easier to read and interpret, despite the large number of arrows (11) entering from the variable level above. Since the activation and inhibition mechanisms are the central explanatory constructs in the model and since one of the stated purposes of the model is to organize the large number of studies in a single framework, I would devote additional prose to referencing those papers which have documented evidence for activation and inhibition (perhaps in footnotes below the model).

Reply: Because this paper does not aim to provide a comprehensive review, we only gave examples for each variable/mechanism. We clarify this in the revised version of the paper (e.g., pp. 1 and 3, and throughout the paper). We added references in the section that discusses activation and inhibition (pp. 9-11).  

Treatment effects. The bottom section of the model makes a tri-partite division of Treatment Effects into positive, null and negative effects with cross-language and within-language effects contained in each module. Six arrows come from the mechanisms above, one to each of the three possible treatment effects. The two strongest effects are depicted by bolded black lines, one from Activation to the positive treatment effects and one from Inhibition to the negative treatment effects. Here I make two suggestions: First, I would separate graphically within-language from cross-language effects, even at the expense of adding more complexity to the model. Second, it would be beneficial to further explicate the nature of the Activation and Inhibition mechanisms that lead to those effects, differentiating those which are within language and cross language.

Reply: The revised paper focuses on cross-language effects and so we removed the within-language effects from our discussion and from the model.

Finally, I offer comments on a number of local issues:

lines 69-73

"Namely, expectations regarding changes associated with treatment would depend on hypotheses regarding the neuronal networks that are associated with the representation and processing of multiple languages in the brain."I would opt for focus on selection and control rather than on representation to sharpen this (see Green&Abutalevi 2008)

Reply: We removed the word “representation”

lines 81-86

"Researchers who put forwardàpropose mechanisms to account for the complex picture of improvement in multilingual PWA draw on what has been proposed in the study of multilingual language processing. Psycholinguistic investigations [add reference] of language performance in multilingual individuals suggest that, in the presence of multiple languages, mechanisms of activation as well as mechanisms of inhibition regulate language use, depending on the communication situation.

Reply: We added a reference there (Green & Abutalebi 2013)

line 102

put forwardàproposed

Reply:  done

line 182

that contributed to a meaningful production out of the total words produced, in a à

that contributed to meaningful production of the total number of words produced in a

Reply: done

line 184

languages with comparably high abilities to the L1,à

languages with high abilities comparable to those in L1,

Reply: done 

line 190

On the one extreme à At one extreme

Reply: done  

line 192

prior to or following the aphasia onset à delete 'the'

Reply: done  

lines 322-328

"As illustrated in sections 2 and 3 above, researchers of treatment in multilingual PWA have reported a mixed pattern of results and have identified critical variables that may influence the results observed. The effects of such variables can be explained by postulating the underlying mechanisms that allow for the languages of a multilingual speaker to facilitate activation of one another as well as to interfere with and inhibit one another. I'm not sure a model which lacks the kind of detail that for example Green's language control model has is sufficient to 'explain' the effects of the variables proposed in the TEAM model. Green 2018 argues that "Varying language activation levels is an insufficient mechanism to explain the variety of language use [in CS]". I recommend a softer term.  

Reply: We changed the wording in that sentence

lines 333-338

"When multiple languages are involved, a given language has to reach a threshold level of activation to be accessible for comprehension and production [5]. The brain lesion resulting in language impairments in multilingual PWA can lead to depressed activation levels in all languages, or, it is possible that only one of the languages reaches the activation threshold at a given time, yielding the patterns of selective impairments that have been reported in the literature [5]."

Reconsider the notion of threshold level of activation; perhaps add a caveat.

 Reply: We added a sentence to qualify the notion of activation threshold (p. 9).

lines 397-399

This complex interaction of a number of such variables can explain the plethora of mixed results reported in the literature on language treatment in multilingual PWA.

Consider changing 'explain' to 'organize,' 'clarify' or something more modest; see comment above.

Reply: We changed the wording in that sentence

Reviewer 2 Report

Summary:

In this article, the authors have reviewed the multilingual aphasia literature focusing on different variables and mechanisms that may impact the efficacy of language treatment in this population. In addition, the authors have proposed an integrated model to capture this complexity.  

I appreciate the authors for selecting such an important topic to review. However, I have few concerns that I feel need to be addressed before accepting for publication and require a major revision.

  1. I would like to know the process of reviewing the literature that is what were the search terms, data bases, years, etc. and based on these criteria how many studies did the author reviewed. A summary table of the literature review would be really useful for the readers.
  2. The authors have selected three multilingualism-related variables (language acquisition, usage, proficiency), three aphasia related variables (lesion site, time post-onset, focus of treatment) and two mechanisms (activation and inhibition) and based on those variables a new model has been proposed. However, throughout the article I could not understand what was the rationale behind selecting only those variables. I think there needs to be a solid theoretical explanation behind selecting those particular variables/mechanisms.
  3. Page 3, line 126-127: the authors cited a reference stating that there may be a difference in the representation and processing of the L1 versus others. The authors need to explain this sentence in more details.
  4. Page 3, line 128-133: the authors talk about differential impairment but again I feel this needs to be explained a bit more. For example, who were the participants in those studies, what languages they spoke, what were the primary assessment methods, etc.
  5. Page 4, the authors talk about language proficiency, language abilities and language dominance and the challenges that are there in the literature to assess these variables. I would like to know what the authors would suggest to overcome those challenges.
  6. Page 4, line 173-187: the authors cite and talk about only one treatment study to address the issue of language proficiency and how that might impact language treatment. Is this the only treatment study to look at the relationship between language proficiency and recovery in multilingual aphasia?
  7. Page 5, line 199: the authors have said that L1 attrition has been studied extensively and cited only one reference. Overall, I think the language attrition section needs to be explained in details.
  8. Page 5, line 235-237: needs citation for this statement.
  9. Page 6, subheading Treatment: the authors mentioned about treatment methods which vary in focus and dose. However, the authors do not go in detail about treatment doses and how that can have an effect on the treatment efficacy.
  10. The authors have proposed a new model at the end. However, the authors do not explain the model. Why the authors believed some variables have direct effect and some have indirect effect. The authors also need to be explicit about how one can test this model otherwise what's new in this model that we already don't know! 

Author Response

We thank the reviewers for their helpful comments. We revised the manuscript according to the reviewers’ suggestions. We simplified our model and added an explanation of the model’s structure. We wish to clarify that we did not aim to conduct a systematic review of the literature, and we now state so in the paper. Our aim was to highlight the variables that have been addressed in the literature on cross-language generalization in aphasia in multilingual people and to propose a model that can serve as a roadmap for researchers so key variables and mechanisms can be considered in tandem. In the revised manuscript, we made sure that the variables discussed in the text align with those included in the model and we added a detailed explanation of the model. We kept all new text in “track changes” as suggested.

Point-by-point response:

In this article, the authors have reviewed the multilingual aphasia literature focusing on different variables and mechanisms that may impact the efficacy of language treatment in this population. In addition, the authors have proposed an integrated model to capture this complexity.  

I appreciate the authors for selecting such an important topic to review. However, I have few concerns that I feel need to be addressed before accepting for publication and require a major revision.

  1. I would like to know the process of reviewing the literature that is what were the search terms, data bases, years, etc. and based on these criteria how many studies did the author reviewed. A summary table of the literature review would be really useful for the readers.

Reply: We would like to clarify that in this paper we did not aim to conduct a systematic review of this literature, but rather to highlight the complexity of the phenomenon and the variability in previous results, and to offer a tool to consider the variables and mechanisms that have been addressed in previous studies. We therefore did not include details about the review process. Indeed, perhaps a review is the wrong term for the paper. We aim here to reflect on the state of the field and to highlight variables and considerations that have been partially addressed in individual previous papers. We now clarify this in the introduction to the paper (pp. 1 and 3)

  1. The authors have selected three multilingualism-related variables (language acquisition, usage, proficiency), three aphasia related variables (lesion site, time post-onset, focus of treatment) and two mechanisms (activation and inhibition) and based on those variables a new model has been proposed. However, throughout the article I could not understand what was the rationale behind selecting only those variables. I think there needs to be a solid theoretical explanation behind selecting those particular variables/mechanisms.

Reply: We appreciate the reviewer’s concern; we have selected these variables as these are the ones that have been discussed in the literature on cross-language generalization and appear most relevant. Here too, we do not mean to be exhaustive but to highlight the discussion in the literature. We clarify this in the revised introduction (pp. 1-3)

  1. Page 3, line 126-127: the authors cited a reference stating that there may be a difference in the representation and processing of the L1 versus others. The authors need to explain this sentence in more details.

Reply: We added an explanation to this statement (p. 4)

  1. Page 3, line 128-133: the authors talk about differential impairment but again I feel this needs to be explained a bit more. For example, who were the participants in those studies, what languages they spoke, what were the primary assessment methods, etc.

Reply: We now clarify earlier in the paper what differential impairment may be and we clarified that the description of the study that follows provides an example for differential response to treatment (p. 4).

  1. Page 4, the authors talk about language proficiency, language abilities and language dominance and the challenges that are there in the literature to assess these variables. I would like to know what the authors would suggest to overcome those challenges.

Reply: We added a comment about ways forward in this regard and a reference to a relevant discussion in another paper (p. 5). A discussion of best practices to overcome these challenges is beyond the scope of this paper.

  1. Page 4, line 173-187: the authors cite and talk about only one treatment study to address the issue of language proficiency and how that might impact language treatment. Is this the only treatment study to look at the relationship between language proficiency and recovery in multilingual aphasia?

Reply: We now clarify that for each of the sections we selected one study as an example. There are over 30 studies in the literature addressing rehabilitation with multilingual speakers; we did not intend to review all of them but rather we selected a handful of studies for the purpose of illustrating each variable we highlight here.

  1. Page 5, line 199: the authors have said that L1 attrition has been studied extensively and cited only one reference. Overall, I think the language attrition section needs to be explained in details.

Reply: We added a short explanation of language attrition as suggested (pp. 4-5).

  1. Page 5, line 235-237: needs citation for this statement.

Reply: This statement represents our own evaluation of the state of the literature.

  1. Page 6, subheading Treatment: the authors mentioned about treatment methods which vary in focus and dose. However, the authors do not go in detail about treatment doses and how that can have an effect on the treatment efficacy.

Reply: We thank the reviewer for pointing this out. We restructured the section on treatment; we highlight the two treatment variables that are most relevant to multilingual people and to cross-language generalization (pp. 8-9).

  1. The authors have proposed a new model at the end. However, the authors do not explain the model. Why the authors believed some variables have direct effect and some have indirect effect. The authors also need to be explicit about how one can test this model otherwise what's new in this model that we already don't know! 

Reply: We added to the explanation of the model and clarified its aim (pp. 12-14), and we now address ways forward to use the model in the field (p. 14).

Reviewer 3 Report

The present paper explores a very relevant topic in the field of aphasia, variation in experimental results, and focuses on which variables may affect recovery outcomes and how can these be integrated in a model (the TEAM) to maximize the effects of the therapy in multilinguals. As such, the paper is a) very relevant and b) very interesting. However, in its present shape, it raises certain concerns.

Bibliography review, Variables and Mechanisms: How have the previous studies been selected among the " substantial literature exploring language treatment effects"? A brief description of the methodology of the exhaustive review would help to have a clearer view of the inclusion criteria.

Related to this, the authors mention " This paper addresses variables and mechanisms that affect efficacy of language treatment " but how were the variables included in the model selected? A justification would also help. Taking section 2 as an example, why focusing on age of acquisition (related to the multilingual status of the participant) vs. for instance frequency or imageability (of the tested items per se) or the proximity of the languages spoken by the multilingual individual? Or, why not including psycho-social factors?

Further details (sample sizes, number of studies for and against, typology of the spoken languages) would make the arguments put forth more robust. Note for instance that in 3.1. after mentioning huge variation across studies the authors conclude: " several treatment studies have proposed evidence for impaired control mechanisms and consequent effects on response to treatment" but mention as an example a single case study: " Abutalebi et al. [48] enrolled a 56-years-old Spanish-Italian bilingual speaker ". In the treatment section, further details would also help clarify the picture as for the potential effects of intervention.

The TEAM model: In its current from, the TEAM stands as a graphical depiction of the text presented above. But how does this model contribute to advancing the field? a) An explanation of the model would be highly welcome, especially considering the forest of arrows present in the capture. b) How can the model be validated? c) How can the predictions of the TEAM be put to work in favor of a better treatment for a specific patient? Note that this is one of the aims of the study: "to aid further study of impairment and rehabilitation in multilingual people with aphasia". d) Or, what does this lead to? How to move forwards from here?

Author Response

We thank the reviewers for their helpful comments. We revised the manuscript according to the reviewers’ suggestions. We simplified our model and added an explanation of the model’s structure. We wish to clarify that we did not aim to conduct a systematic review of the literature, and we now state so in the paper. Our aim was to highlight the variables that have been addressed in the literature on cross-language generalization in aphasia in multilingual people and to propose a model that can serve as a roadmap for researchers so key variables and mechanisms can be considered in tandem. In the revised manuscript, we made sure that the variables discussed in the text align with those included in the model and we added a detailed explanation of the model. We kept all new text in “track changes” as suggested.

Point-by-point response:

The present paper explores a very relevant topic in the field of aphasia, variation in experimental results, and focuses on which variables may affect recovery outcomes and how can these be integrated in a model (the TEAM) to maximize the effects of the therapy in multilinguals. As such, the paper is a) very relevant and b) very interesting. However, in its present shape, it raises certain concerns.

Bibliography review, Variables and Mechanisms: How have the previous studies been selected among the " substantial literature exploring language treatment effects"? A brief description of the methodology of the exhaustive review would help to have a clearer view of the inclusion criteria.

Reply: We now explain that we did not aim to conduct a systematic nor exhaustive review of the literature but rather to highlight and reflect on main issues and the key variables that have been raised in the literature on cross-language generalization (pp. 1 and 3).

Related to this, the authors mention " This paper addresses variables and mechanisms that affect efficacy of language treatment " but how were the variables included in the model selected? A justification would also help. Taking section 2 as an example, why focusing on age of acquisition (related to the multilingual status of the participant) vs. for instance frequency or imageability (of the tested items per se) or the proximity of the languages spoken by the multilingual individual? Or, why not including psycho-social factors?

Reply: We appreciate this concern and added a rationale for the selection of the variables we include (p. 3). We focus on variables that have been raised in the context of cross-language generalization. We now added a mention of the variables raised by the reviewer (p. 8).    

Further details (sample sizes, number of studies for and against, typology of the spoken languages) would make the arguments put forth more robust. Note for instance that in 3.1. after mentioning huge variation across studies the authors conclude: " several treatment studies have proposed evidence for impaired control mechanisms and consequent effects on response to treatment" but mention as an example a single case study: " Abutalebi et al. [48] enrolled a 56-years-old Spanish-Italian bilingual speaker ". In the treatment section, further details would also help clarify the picture as for the potential effects of intervention.

Reply: Considering the scope of this paper, we did not aim to review all the papers in the literature (over 30) that address cross-language generalization; rather, we selected individual examples to highlight the issues we reflected on. We hope we clarified this in the revised paper.  

The TEAM model: In its current from, the TEAM stands as a graphical depiction of the text presented above. But how does this model contribute to advancing the field? a) An explanation of the model would be highly welcome, especially considering the forest of arrows present in the capture. b) How can the model be validated? c) How can the predictions of the TEAM be put to work in favor of a better treatment for a specific patient? Note that this is one of the aims of the study: "to aid further study of impairment and rehabilitation in multilingual people with aphasia". d) Or, what does this lead to? How to move forwards from here?

Reply: In the revised draft of the paper, we tried to clarify the aim of the model. We hope that it helps bring together the variables and mechanisms that had been discussed in the literature (in a more disjointed manner) and to serve as a road map for researchers and clinicians as they consider treatment and cross-language effects in multilingual people with aphasia. At this preliminary stage, no clear predictions can be derived from the model. Future studies can shed additional light on the interaction among the variables, systematic reviews can help understand current trends in cross-language results, and computational models can help predict the relative weight of each variable. We added a discussion of these points in the last section of the paper (pp. 12-14)

Round 2

Reviewer 2 Report

I am happy with the corrections made and would recommend for publication. Congratulations to the authors. 

Reviewer 3 Report

The authors have clearly addressed the issues raised in the previous review and provided a clear explanation for the decisions taken thus justifying the acceptance of the paper in its present form.